# Generative AI and academic scientists in US universities: Perception, experience, and adoption intentions

Wenceslao Arroyo-Machado[1]*, Jinghuan Ma[1], Tipeng Chen[1], Timothy P. Johnson[1,2], Shaika Islam[1], Lesley Michalegko[1], Eric Welch[1]

1 Center for Science, Technology and Environmental Policy Studies, School of Public Affairs, Arizona State University, Phoenix, Arizona, United States of America, 2 Department of Public Policy, Management, and Analytics, University of Illinois at Chicago, Chicago, Illinois, United States of America

* warroyom@asu.edu

## Abstract

The integration of generative Artificial Intelligence (AI) into academia has sparked interest and debate among academic scientists. This paper explores the early adoption and perceptions of US academic scientists regarding the use of generative AI in teaching and research activities. To do so, this analysis focuses exclusively on STEM fields due to their high exposure to rapid technological advancements. Drawing from a nationally representative survey of 232 respondents, we examine academic scientists' attitudes, experiences, and intentions regarding AI adoption. Results indicate that 65% of respondents have utilized generative AI in teaching or research activities, with 20% applying it in both areas. Among those currently using AI, 84% intend to continue its application, indicating a high level of confidence in its perceived benefits. AI is most frequently used in teaching to develop pedagogical materials (51%) and in research for writing, reviewing, and editing tasks (40%). Despite concerns about misinformation, with 78% of respondents indicating it as their top concern regarding AI, there is broad recognition of AI's potential impact on society. Most academic scientists have already integrated AI into their academic activities, demonstrating cautious yet optimistic adoption due to perceived risks. Furthermore, there is strong support for academic-led regulation of AI, highlighting the need for responsible governance to maximize benefits while minimizing risks in educational and research settings.

## Introduction

Research in Artificial Intelligence (AI) has a storied past, beginning when the term was first coined at the 1956 Dartmouth Summer Research Project on Artificial Intelligence [1]. Fast forward to late 2022, the release of ChatGPT (https://openai.com/blog/chatgpt) marked a significant milestone in AI, democratizing access to advanced generative AI technologies. Generative AI refers to a subset of artificial intelligence

**Data availability statement:** The dataset from the study, which includes the complete set of survey questions and corresponding answer options, is available at the Roper Center for Public Opinion Research at Cornell University (https://doi.org/10.25940/ROPER-31122347).

**Funding:** The author(s) received no specific funding for this work.

**Competing interests:** The authors have declared that no competing interests exist.

that is designed to create new content, such as text, images, audio, video, and even code, based on patterns learned from large datasets [2]. Unlike traditional AI systems, which function based on pre-defined rules, generative AI systems demonstrate greater functional autonomy and ability to perform intelligent behaviors in a way that mimic human creativity by learning patterns and behaviors from data [3]. The rise of generative AI, particularly through chatbots powered by including Large Language Models (LLM), further exemplifies the impact of these technologies on various domains. The rapid development of chatbots beyond ChatGPT has sparked considerable interest [4]. The widespread adoption of these AI tools has the potential to alter numerous elements of society, including impacting higher education and scientific research [5,6]. The ability of generative AI to produce text that is virtually indistinguishable from human-written work promises to transform educational methodologies and research practices, offering new opportunities for institutions, educators, and students [7,8]. This rapid expansion has also raised important questions regarding risks such as bias, transparency, and social impact [9].

The significance of generative AI's impact is particularly evident in Science, Technology, Engineering, and Mathematics (STEM) fields. The importance of these fields is underscored by the growing number of students pursuing STEM disciplines, a trend influenced by various factors and reflective of a country's economic development [10]. Simultaneously, since 2015, US companies have been increasing their demand for AI-related expertise and are recruiting early-career AI researchers from academia [11,12]. This growth and demand highlight the importance of integrating AI into educational curricula and research methodologies to keep pace with technological advancements and opportunities. Understanding their perspectives on and experiences with AI is essential, as these academic scientists can provide valuable technical input for its ongoing development across various disciplines [13].

Given the rapid global adoption of generative AI, this article explores the early adoption behaviors and perceptions of US academic scientists related to teaching and research activities. According to innovation diffusion theory [14], early adopters are different from late adopters as they often face higher uncertainty and different motivations compared to later adopters. In this context, early adoption refers to the initial phase of AI integration before it becomes widespread in academia. While the study is centered on the United States, the insights can be considered reflective of broader international trends due to the influential role of US academia on a global scale [15]. To achieve these insights, we draw from a nationally representative survey of academic scientists, that include assistant, associate and full professors, across six disciplines—biology, chemistry, computer and information science engineering, civil and environmental engineering, geography, and public health—at randomly selected Carnegie-designated Research Extensive and Intensive (R1) universities. The study is designed as a descriptive and exploratory analysis of academic scientists' reported attitudes, experiences, and uses of generative AI. The following aims were established to effectively address this main objective:

1. To assess the general attitudes of academic scientists towards generative AI.

2. To explore the perceptions and experiences of academic scientists regarding the use of AI in academic teaching and research.

3. To examine how academic scientists' adoption intention for teaching and research varies.

4. To investigate academic scientists concerns and perceptions regarding the regulation of generative AI.

The next section presents a literature on the use of AI in higher education. It explores how academic scientists from different disciplines perceive its application and reviews other survey-based research on this topic. The subsequent methodology section describes the methods and data collection process is followed by a results section that presents the findings of the survey, detailing academic scientists' perceptions, experiences, and intentions regarding generative AI in teaching and research. The conclusion section synthesizes key insights and points out limitations of the study.

### Literature review

The following review is structured around the benefits and risks of generative AI in academia, setting the stage for an exploratory analysis of scientists' behaviors and perceptions.

**Generative AI in academic education.** Artificial intelligence (AI) has long been an integral part of higher education at different levels, significantly predating the recent explosion in AI technologies such as ChatGPT, Microsoft Copilot or Elicit [16]. In its early applications in this context, AI was employed to streamline complex administrative tasks, enhancing efficiency in areas like admissions processes and predicting student retention rates, which are crucial for institutional planning [17,18]. Additionally, academic libraries began to leverage AI for automated cataloging, intelligent search tools, AI-driven reference services, and adaptive learning systems, improving accessibility and operational efficiency [19].

More recently, generative AI tools has introduced new possibilities and challenges across various domains in higher education [20]. AI is used to personalize learning through adaptive tutoring systems and identify and address student learning obstacles [21,22]. Advocates suggest that generative AI-enhanced personalized education could improve learning outcomes, increase accessibility, reduce inequities [23–26], and enhance student interaction and engagement [27–29]. Moreover, AI is increasingly used to assist in assessment processes, making them more accurate and less time-consuming, and enabling automated grading that involves text generation [22,24,25]. Although these applications have relied on specialized and custom-designed applications that are not universally accessible, they are beginning to transform the higher education system by offering institutions and individuals substantial administrative and educational benefits. Educators recognize that the impact on education will vary by discipline and require changes in teaching methods [30].

Despite the perceived usefulness and improved ease of communication [31], there is insufficient understanding of generative AI's long-term impacts on educational practices, particularly regarding sustained changes in teaching methods and learning outcomes [32], and of how academic scientists perceive and respond to these developments. These uncertainties are further complicated by significant, pre-existing concerns regarding the reliability and safety of AI systems in education, issues which have broadened and intensified following generative AI's widespread adoption [33–35]. Notable issues include the risk of overreliance on AI and its effects on critical thinking [36,37] and AI's potential to undermine creativity, hinder academic integrity, and reinforce existing inequalities due to biases in datasets, algorithmic opacity, and disparities in access to AI-driven tools [38]. AI's potential to undermine creativity, hinder academic integrity, and exacerbate existing inequalities is reflected in disparities in access to AI tools and the reinforcement of biases within algorithmic decision-making. Generative AI can also compromise academic standards through plagiarism [26] and generate misleading or fabricated information, thereby undermining trust [39]. Concerns also remain about fairness, responsibility, and the necessity for clear guidelines and training to ensure generative AI's ethical and effective implementation [40,41]. While AI integration offers benefits like improved accessibility and efficiency, it also raises fears about job obsolescence due to the evolving demand for new skills and competencies [42]. These concerns point to a growing need for human oversight to safeguard

academic integrity and mitigate bias [36,43], as the adoption of generative AI in education raises significant challenges related to critical thinking, fairness, and equitable access to learning opportunities.

**Generative AI in academic research.** Generative AI is also having a substantial impact on academic research. Messeri and Crockett [44] have outlined a typology of four different roles generative AI can play in research: oracle for study design, surrogate for data collection, quant for data analysis, and arbiter for peer review. Generative AI tools have sparked interest for their potential in various tasks along the research pipeline of discovery by facilitating the synthesis and understanding of scientific literature, generating and organizing ideas, providing targeted feedback, and analyzing vast amounts of data to uncover connections, thus streamlining research processes and mitigating some types of cognitive biases [45]. Additionally, AI models can improve the prediction of future discoveries, especially in areas with limited literature, by anticipating human predictions and identifying key researchers [46] or by acting as "an autonomous researcher," to produce new knowledge [47].

While many of the applications of generative AI to research activities appear promising, challenges and risks still loom large [5,48,49]. For example, the expansion of Large Language Models for broader use presents significant threats to research integrity, impacting its transparency, reliability, and authenticity [49–51]. This can be seen in how AI systems can introduce biases from their training data, potentially perpetuating hidden prejudices and undermining the integrity and authenticity of scientific knowledge [52]. As universities start to craft generative AI guidelines [53], academic scientists' excitement about potential efficiency gains is balanced by concerns over risks such as misinformation, plagiarism, and falsified research [48,54]. For example, while generative AI can produce high-quality text suitable for publication, its limited capacity for research design and data analysis increases the risk of misinformation, fabricated findings, and academic integrity violations [55]. Moreover, concerns have been raised about who owns the AI-generated content [56], whether it maintains its integrity [57], and what the risks are to data privacy when sharing protected information [58]. These issues pose serious challenges that could affect the perceptions and behaviors of academic scientists leading to caution in embracing these technologies and potentially slowing down their integration into research practices.

**Academic perceptions about and adoption of generative AI.** Understanding academic scientists' perceptions of generative AI is crucial because of its potential to radically transform research and teaching landscapes in higher education, affecting vast segments of the US and international academic communities. Conceptual frameworks such as the Theory of Reasoned Action (TRA) [59], the Theory of Planned Behavior (TPB) [60], the Technology Acceptance Model (TAM) [61], the Innovation Diffusion Theory [62] and the Unified Theory of Acceptance and Use of Technology (UTAUT) [63] provide valuable frameworks for examining technology acceptance. Cumulatively, they highlight the importance of personal beliefs and attitudes, including concerns [64] confidence and readiness [61,65], as well as past experiences [65] in shaping technology acceptance. However, the implications of AI extend beyond adoption frameworks; AI's integration into academic environments presents unprecedented opportunities and significant uncertainties, making it imperative to closely monitor how these changes unfold over time.

While educators and researchers recognize AI as a valuable tool for developing new ideas, streamlining workflows, and offering remote and personalized learning experiences [66–68], they have also expressed ethical concerns regarding plagiarism, misinformation, and misidentification of AI-produced content as human work. The challenge of accurately detecting AI-generated text is significant, as current classifiers often misclassify human-written text as generated by AI and vice versa [69,70]. These risks affect various fields differently. For instance, in medical education, while there is optimism about AI's potential to reduce administrative burdens and enhance diagnostic accuracy, concerns persist about data privacy, the potential for AI-generated medical content to be overly relied upon without human oversight, and the challenges of ensuring the accuracy and reliability of AI-assisted educational tools [71]. Researchers in population health and academic publishing have also voiced concerns over sensationalist media portrayals of generative AI, stressing the importance of better education and awareness raising to combat these misconceptions [72,73]. Additionally, AI and machine learning specialists have identified a need for more safety research and exhibit low levels of trust in certain organizations managing AI

technologies [74]. Given the complex and uncertain consequences of AI integration into academia, systematic tracking and continuous evaluation of this transformation are essential to responsibly navigate the potential benefits and risks.

With its significant promise for advancing education and research, generative AI presents both compelling opportunities and serious concerns. Understanding how academic scientists weigh these perceived benefits against potential risks is essential for assessing whether adoption is likely to accelerate, slow down, or proceed cautiously. However, few studies have jointly addressed STEM academic scientists' perceptions of, experiences with, and generative AI adoption intentions in both education and research. In order to bridge this gap, this study develops a comprehensive approach to better delineate the current state and more accurately understand the adoption rationales of academic scientists. We believe such research is a prerequisite for tailoring educational methodologies, enhancing research practices, and addressing concerns regarding the regulation of generative AI.

## Materials and methods

We make use of a unique database consisting of two sets of merged survey data from the SciOPS (Scientist Opinion Panel Survey): an intake survey administered in the Spring of 2022 and 2023 to collect general socio-demographic data regarding SciOPS panelists, and another survey of panelists' attitudes towards AI that was administered in late 2023 (hereafter, the AI survey). SciOPS is a platform to improve science communication between university academic scientists and the public by understanding scientist opinions on current topics relevant to the science community and to society. Detailed information regarding SciOPS and past surveys administrated by this platform can be found at SciOPS website [75].

This study employed a two-stage sampling design. At the first stage, SciOPS panel members were recruited from a full sample frame that includes approximately 18,500 PhD-level academic scientists employed by Carnegie-designated Research Extensive and Intensive (R1) universities across the US, representing six STEM fields: biology, chemistry, civil and environmental engineering, computer and information science engineering, geography, and public health. The SciOPS research team used probability sampling to randomly select R1 universities and collected the names and contact information of tenured and tenure track faculty (i.e., assistant, associate, and full professors), and non-tenure track researchers with PhDs from each sampled academic department website. Fields and ranks were collected by the SciOPS research team from their personal profile on their institution websites. Table S3 in S2 Appendix shows the number of randomly selected institutions by field.

A total of 1,365 eligible academic scientists consented to become SciOPS panel members, with a recruitment rate of 7.5%, calculated according to the guidance of American Association for Public Opinion Research (AAPOR) Recruitment Rate formula [76]. The intake survey was dispensed to all 1,365 members to capture their general socio-demographic information, including birth year, self-reported gender, citizenship, academic rank, and academic discipline.

In the second stage, 777 SciOPS panel members were randomly selected from all members and invited to participate in this survey. To construct valid survey questions, the SciOPS research team designed them based on literature on technology adoption among academic scientists. The questions underwent multiple rounds of review, revision, and internal pretesting by 10 SciOPS research team members. The full questionnaire is available in S3 Appendix.

The survey was administrated online in English through the Nubis® system, which is an online software platform for administering questionnaires with the aim of protecting the confidentiality of survey respondents. A pre-notification electronic email was delivered on September 27, 2023 to notify the sample of a forthcoming survey invitation. We sent email messages with a formal survey invitation to each sampled individual along with a questionnaire hyperlink on September 29. Three formal reminder messages were sent on October 4, October 11, and October 18. A final short appeal message was sent on November 2. We closed the survey on November 4. This survey obtained a total of 232 usable responses, representing an individual survey completion rate (COMR) of 29.9% (RR4) and an AAPOR Cumulative Response Rate (CUMRR) of 2.2%. The CUMRR calculated by multiplying the RECR for the SciOPS panel (7.5%) with the Completion Rate for this survey (29.9%) and dividing by 100.

Both surveys were approved by the Institutional Review Board at Arizona State University (IRB Approval No. STUDY00012476). Data were weighted to account for probabilities of selection and post-stratified by gender, academic field, and rank to represent the population of academic scientists from which the full sample was initially recruited. The margin of sampling error for the AI survey estimates is ± 6.4 percentage points, based on a population of 18,504 and a 95% confidence level. Table 1 reports the weighted demographic characteristics of survey respondents.

Additionally, to assess how variations in question framing might influence academic scientists' attitudes toward government regulation of generative AI, we embedded an experimental design within the survey. Respondents were randomly assigned to one of four versions of a key question, which varied in terms of the inclusion of a moderate response option and the presence or absence of ethical framing regarding potential risks. This design allowed us to examine how different presentations of the same regulatory dilemma influenced participants' preferences.

To assess the representativeness of our sample, we also conducted a non-response bias analysis (see S1 Appendix for details). The analysis shows minimal differences between the completed sample and the initial sample frame of academic scientists, and no significant differences between our sample and the panel members invited to participate in the survey. These findings confirm that the dataset provides a reasonable representation of US-based academic scientists.

## Results

Following, we detail survey results on academic scientists' perceptions of generative AI use and regulation, underscoring their experiences and intentions regarding its implementation in teaching and research.

### General perceptions and practical use of generative AI in academia

Survey responses indicate that a majority of academic scientists (93%) believe that generative AI will have a significant impact on daily life in the United States (Fig 1). The potential effects of climate change also register as a major point of academic concern, albeit slightly less so (88%) than generative AI (though this difference falls within the margin of error),

**Table 1. Weighted descriptive statistics of the final sample.**

| Construct | Variable | N | % |
|---|---|---|---|
| Self-reported Gender | Female | 71 | 31 |
| | Male | 155 | 66 |
| | Prefer not to report | 6 | 3 |
| Field | Biology | 16 | 7 |
| | Chemistry | 31 | 13 |
| | Computer and Information Science Engineering | 28 | 12 |
| | Civil and Environmental Engineering | 63 | 27 |
| | Geography | 90 | 39 |
| | Public Health | 4 | 2 |
| Rank | Full Professor | 76 | 33 |
| | Associate Professor | 49 | 21 |
| | Assistant Professor | 49 | 21 |
| | Non-tenure Track Researcher | 58 | 25 |
| Self-reported Citizenship | Born a U.S. Citizen | 135 | 58 |
| | Naturalized U.S. Citizen | 60 | 26 |
| | Non-U.S. citizen with a permanent U.S. resident visa (e.g., green card) | 26 | 11 |
| | Non-U.S. citizen with a temporary U.S. resident visa | 4 | 2 |
| | Prefer not to answer | 7 | 3 |

indicating a sense of urgency regarding environmental shifts. While global conflicts (78%), the proliferation of infectious diseases (77%), and the challenges of resource scarcity (72%) are acknowledged, they are perceived as less immediately impactful compared to technological and political shifts on the horizon. Notably, fusion energy, while revolutionary in concept, is seen by academic scientists as trailing behind in terms of imminent impact.

Fig 2 presents academic scientists' specific concerns regarding this technology. Foremost among these is concern over misinformation, which echoes the national discourse on political polarization, with 78% of respondents marking it as extremely or very concerning. Concerns extend to an over-dependence on AI technologies and their potential cybersecurity risks, noted by 60% and 51% of academic scientists, respectively, as major issues; which is followed by the fear that generative AI may stifle human creativity, with half of the surveyed academic scientists expressing apprehension. In contrast, issues such as privacy loss and job displacement, though acknowledged, elicit less concern, where 44% and 24% of respondents express extremely or very strong concern respectively.

Despite these concerns, there is a noteworthy trend among academic scientists integrating generative AI into their core activities: teaching and research. A sizable 65% of academic scientists have personally used generative AI to some extent. Specifically, 40% of surveyed academic scientists report using generative AI in teaching and research, demonstrating an early adoption—defined here as the integration of generative AI into academic activities shortly (one year) after the release of ChatGPT-3.5 in late 2022. In contrast, 25% utilize generative AI for alternative functions not directly related to their primary roles, and 35% have not engaged with AI personally. Diving deeper, 20% of academic scientists report using generative AI in both teaching and research. This is juxtaposed with 11% who use generative AI exclusively for research and a smaller 9% who apply it solely to teaching.

### Irregular use of generative AI for teaching with persistent continuation for adopters

In terms of its adoption for education, generative AI is experiencing early-stage integration by academic scientists in teaching roles, with usage patterns reflecting cautious exploration rather than full integration (Fig 3). The data shows a majority engage with generative AI tools occasionally, reserving them for specific tasks rather than regular use. This tentative approach suggests an awareness of the potential and limitations of current generative AI technologies in educational settings. For those integrating generative AI into their pedagogy, its most common application lies in the development

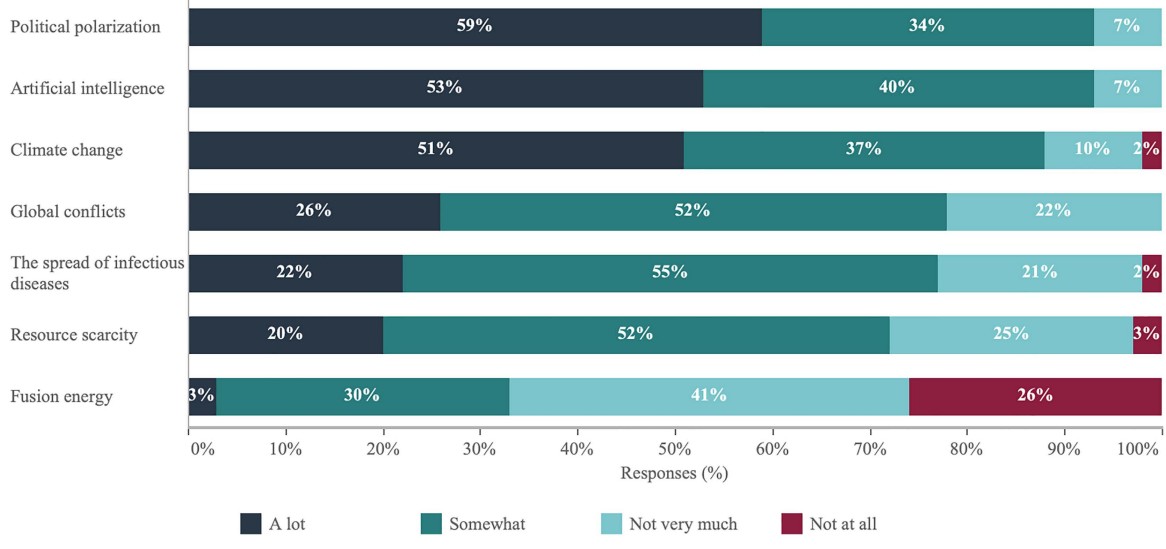

**Fig 1. Academic scientists' views on the extent to which different issues can cause changes to the way we live in the U.S.**

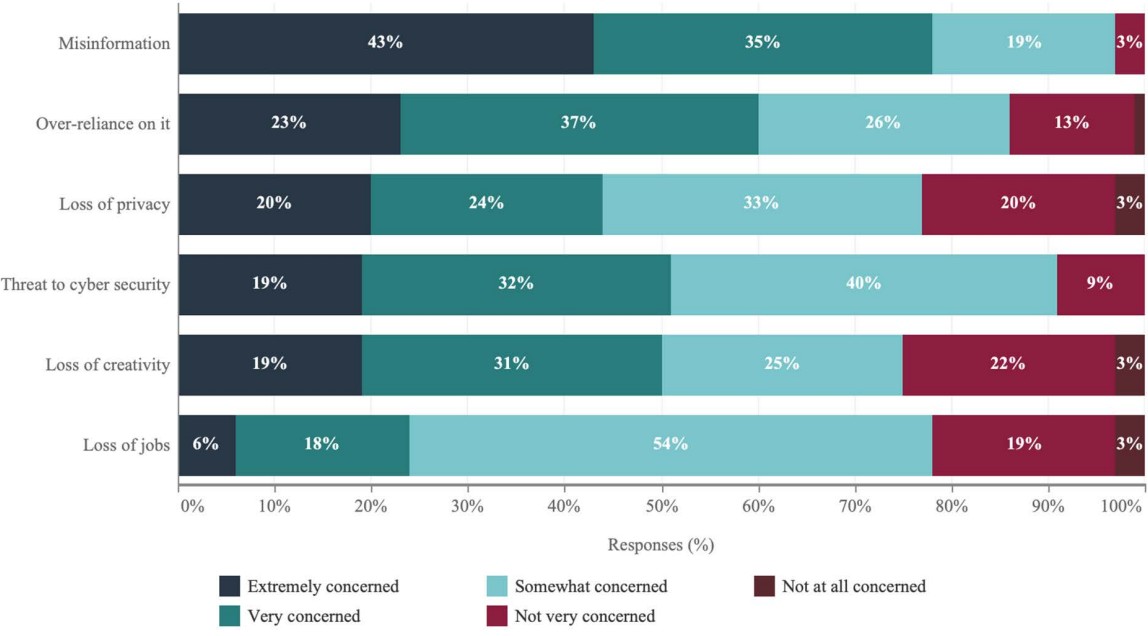

**Fig 2. Academic scientists' level of concern over potential threats posed by generative AI.**

of teaching materials, with 51% of the generative AI-utilizing academic scientists relying on these tools for this purpose. Generative AI is also employed in crafting student assignments (38%) and conducting interactive classroom activities (37%), though less frequently for direct student interactions such as mentoring (17%) or critical tasks like grading (11%) or examinations (8%). The restrained use in these areas may point to a preference for human oversight where subjective judgment and personalized feedback are valued.

Regarding future intentions to use generative AI in teaching, the distinction between current users and non-users is stark. Almost all academic scientists who have integrated generative AI into their teaching methods, 97%, anticipate continuing its use, underscoring a high level of satisfaction and perceived value. In contrast, among those who have not yet employed generative AI in their teaching practice, skepticism is prevalent. A majority, 80%, express reservations about adopting generative AI: within this group, 45% express no intention to use it, and 35% remain undecided. Only 20% of current non-users express openness to future use; within this group, 85% (i.e., 17% of the total) state they are likely to adopt it, while 15% (i.e., 3% of the total) are certain they will incorporate it into their teaching.

## Same perspectives and intentions for generative AI use in research

In the realm of research, the adoption of generative AI exhibits trends similar to its use in educational activities (Fig 4). Nearly half (46%) of the researchers surveyed employ generative AI less than once a month, indicating a tentative exploration, and about a quarter (24%) use it on a weekly basis. Additionally, among academic scientists who only use generative AI in research, 40% report finding value in its applications for writing, reviewing, and editing. Significantly, generative AI's role in data analysis (32%) and especially in research conceptualization (28%) marks a shift, indicating its potential not just in routine tasks but in critical thinking and idea generation phases. Moreover, generative AI's application extends to administrative responsibilities like funding acquisition, demonstrating its utility in streamlining research workflows and relieving administrative burden. Additionally, its application in visualization and other specialized tasks further highlights generative AI's versatility and usefulness in the research process.

## (a) Frequency of use of generative AI for teaching activities

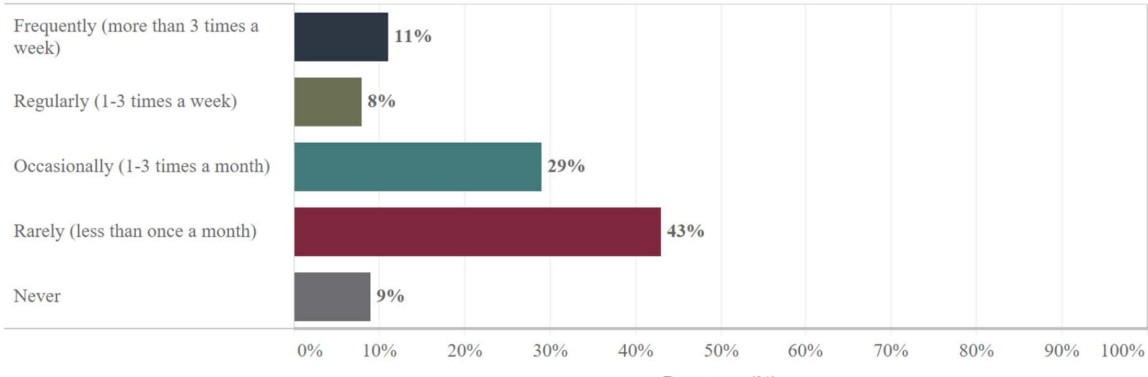

## (b) Use of generative AI for teaching activities

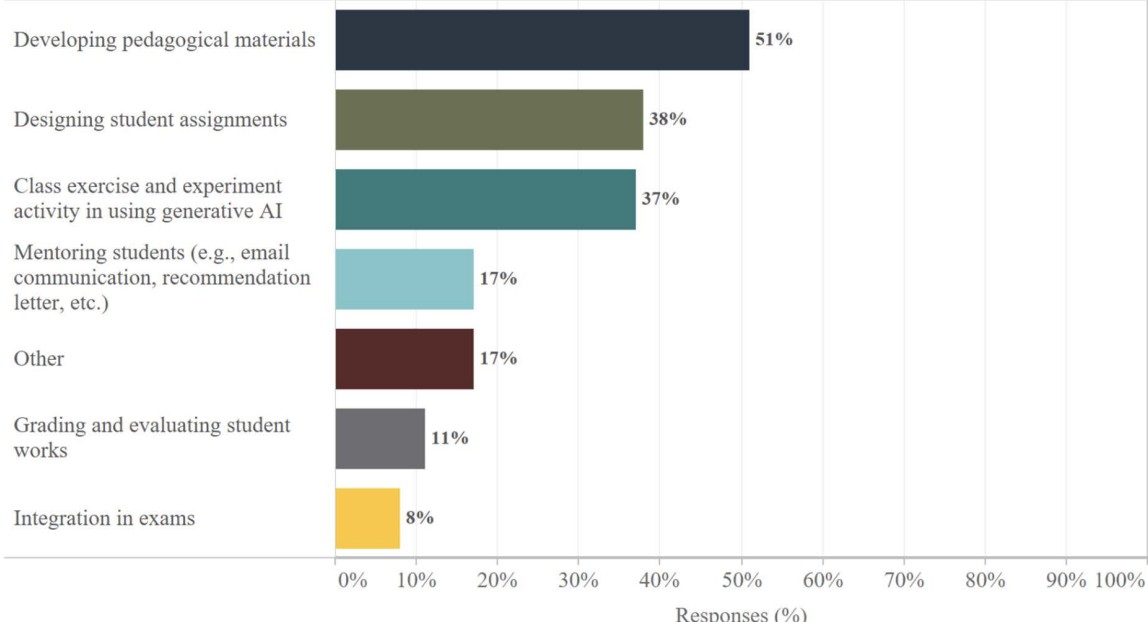

**Fig 3. Frequency and academic scientists use of generative AI for teaching activities.**

Building on the observed patterns of generative AI use in research, future intentions among academic scientists also further highlight the divide between current users and non-users. Those with experience using generative AI in their research overwhelmingly endorse its continued application, with 84% affirming they will likely or definitely continue integrating generative AI into their work. This strong positive sentiment contrasts sharply with the skepticism seen among those yet to adopt generative AI for research, where 78% express reservations. Within this group, a notable 41% are decided against using generative AI, and 37% remain uncertain about its potential benefits. Conversely, only a minority are open to the possibility, with 18% leaning towards probable use and a mere 5% convinced of its definite future use. This dichotomy suggests that firsthand experience with generative AI may influence perceptions of its utility, with a divide emerging between adopters and skeptics regarding its role in research activities.

## (a) Frequency of use of generative AI for research activities

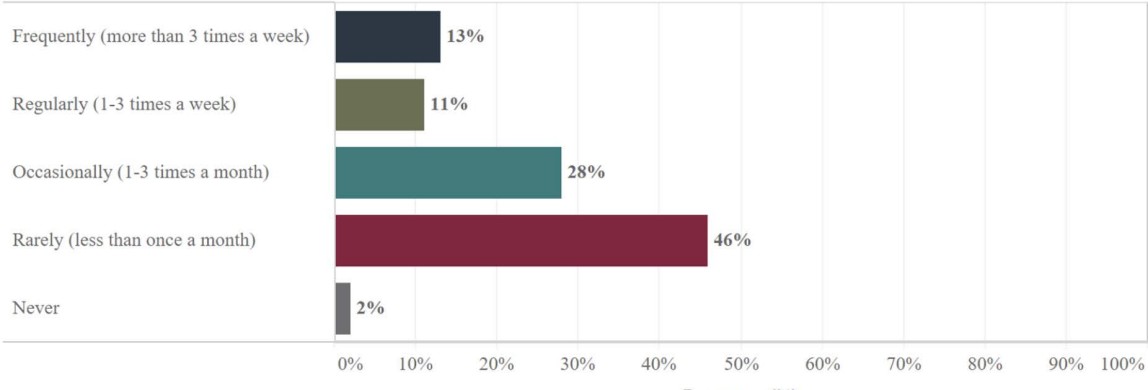

## (b) Use of generative AI for research activities

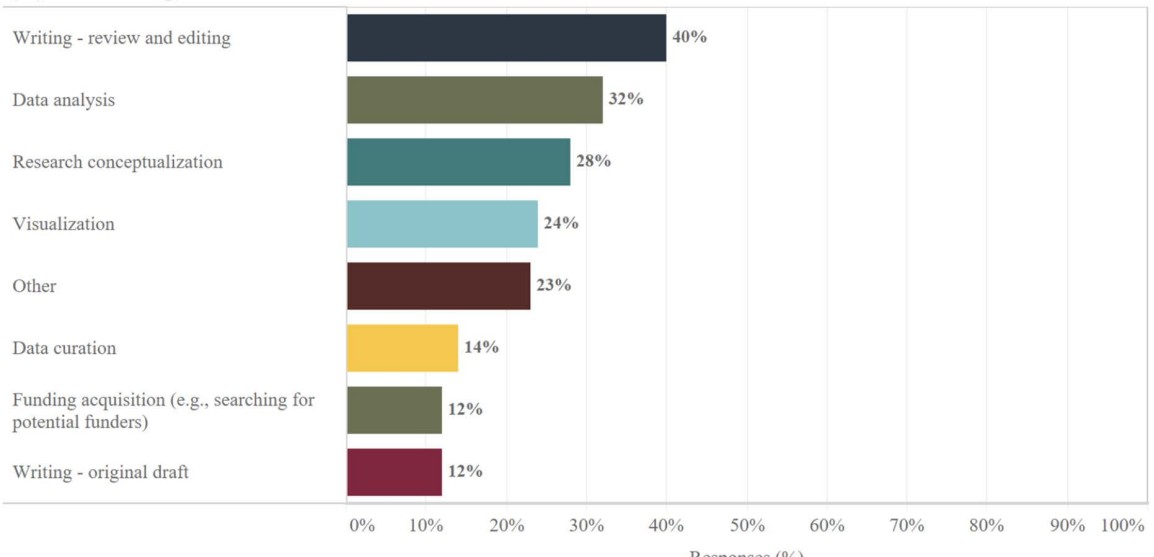

**Fig 4. Frequency and academic scientists use of generative AI for research activities.**

### The university as main regulatory agent and concerns about its regulation

Following the exploration of academic scientists' perspectives and their usage of generative AI, attention shifts to the question of regulatory oversight. Within this domain, there emerges a clear consensus favoring academic institutions as the primary regulators of generative AI. A majority, 88% of academic scientists, underscore the pivotal role that universities should assume, distinguishing them from other key players. National professional associations, journal editors, and publishers also receive notable endorsement, with 74%, 65%, and 61% support respectively, highlighting the broader academic community's involvement in regulation. In contrast, when it comes to entities outside the academic sphere, over half of all academic scientists believe that both the Federal Government (56%) and supranational organizations (54%) should play substantial roles. This distribution of responsibilities suggests a preference for an academic-led approach to generative AI regulation, complemented by governmental and international oversight, to effectively address the intricacies of generative AI's integration into academic settings. Building on this preference for an academia-centered regulatory

approach, there is also strong support among academic scientists for federal involvement in the oversight of generative AI's use, with 71% expressing strong or somewhat strong approval. Alongside the preference for an academia-centered regulatory approach, 71% of academic scientists express strong or somewhat strong approval for federal involvement in generative AI oversight.

To assess how variations in question wording might influence academic scientists' attitudes toward government regulation of generative AI, respondents were randomly assigned to one of four experimental versions of a key survey question: (1) "Which of the following is closest to your current point of view about the role of the government in regulating generative AI?" with responses: "Federal government should not regulate generative AI technology, leaving its regulation to private industry", or "Federal government should ban generative AI deployment until comprehensive research on its potential consequences is conducted"; (2) identical wording but including a moderate response option: "Federal government should regulate generative AI technology, however, it should momentarily step back from doing so until comprehensive research on its potential consequences is conducted"; (3) explicit ethical framing: "Considering the possible ethical and unintended consequences generative AI may have, which of the following is closest to your current point of view about the role of the government in regulating generative AI?" with the same two responses as version 1; and (4) identical ethical framing as version 3, plus the moderate response option included in version 2.

The survey went a step further to dissect academic scientists' attitudes towards government regulation of generative AI through an experimental approach (Fig 5). This segment aimed to evaluate how the phrasing of questions, particularly those mentioning the ethical and unintended consequences of generative AI, alongside the option of a neutral response, would affect their attitudes for federal regulation. Participants were randomly assigned one of four variations of the question, covering aspects such as outright banning versus not banning generative AI technologies and the inclusion or exclusion of a neutral middle response category. This experiment indicates that the question format in which response options were structured significantly influenced participants' responses. Through this detailed exploration, it became apparent that academic scientists' opinions on generative AI governance are complex, with a considerable number supporting some level of federal oversight, yet their preferences are nuanced, often swayed by the contextual presentation of potential generative AI risks and regulatory options.

Our analysis reveals academic scientists' nuanced preferences regarding government regulation of generative AI technologies (Fig 5). When presented with a middle-ground response option about the extent of governmental oversight, a clear majority favored this moderate stance (76% and 59% for two different question versions), demonstrating a shift away from more polarized positions. Specifically, the inclination to outright ban generative AI dropped dramatically (from 50% to 8% and 55% to 28% across the two versions) when this middle option was available. Similarly, the preference for no regulation also decreased (from 44% to 14% and 38% to 9%), indicating a general consensus towards a balanced regulatory approach. Moreover, when questions explicitly highlighted the potential threats posed by generative AI, there was a noticeable increase in support for temporary bans by the Federal Government until thorough research is completed, with or without a middle response option. This ranged from 50% to 55% without a middle option and jumped from 8% to 28% with it. Additionally, the absence of a moderate option slightly increased the likelihood of respondents choosing not to answer (from 2% to 6%, and from 4% to 7% in the respective question versions). These findings underscore the complexity of academic scientists' attitudes towards generative AI regulation, leaning towards a preference for moderate, well-considered policy approaches that balance potential risks and benefits while awaiting further research.

## Discussion

The emergence of ChatGPT stimulates new discussion on the adoption of AI-assisted technology in different sectors. This paper contributes to the understanding of university academic scientists' perceptions of the use and regulation of generative AI, as well as their experience using it in teaching and research activities. Drawing from a nationally representative survey of 232 respondents from STEM disciplines, the study reveals that 65% of academic scientists have incorporated

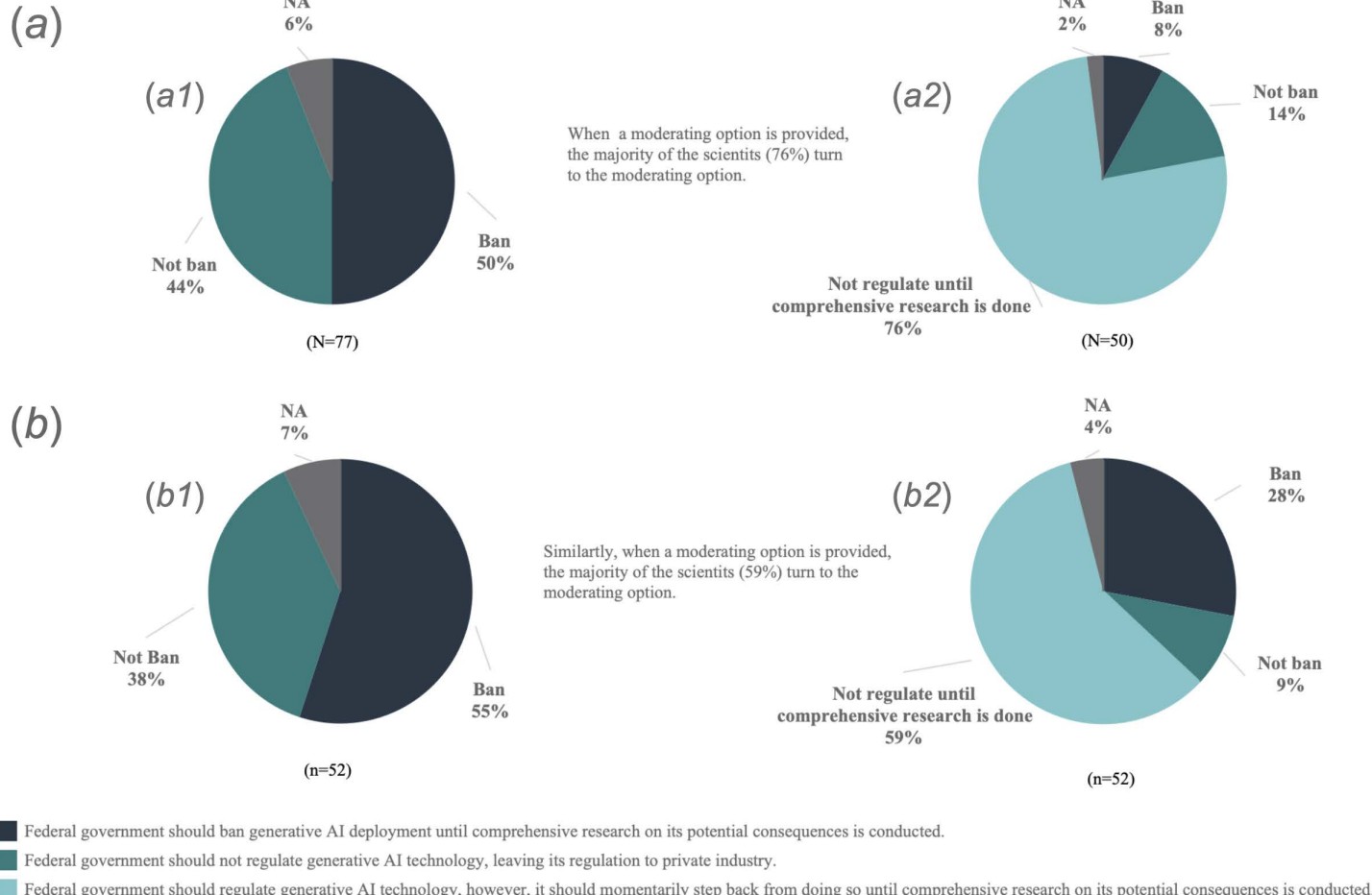

**Fig 5. Methodological experiment results: (a) Current opinion about the role of the government in regulating generative AI; (b) Current opinion about the role of the government in regulating generative AI, considering the possible ethical and unintended consequences it may have.** Confidence intervals and sample sizes for each subsample are as follows: (a1) n = 77, CI = ±11.1%; (a2) n = 50, CI = ±13.8%; and (b1 and b2) n = 52, CI = ±13.6%.

generative AI into teaching or research, with 20% actively using it in both fields. Generative AI is predominantly used in educational settings for developing pedagogical materials (51%) and in research primarily for writing, reviewing, and editing tasks (40%). Despite widespread adoption, concerns persist, particularly about misinformation, identified by 78% of respondents as their primary apprehension. The study further indicates a cautious yet optimistic approach among scientists, with 97% of teaching users and 84% of research users intending to continue its use, demonstrating substantial confidence in its potential benefits. Moreover, regulatory attitudes reveal a pronounced preference (88%) for academic-led oversight of AI integration, reflecting skepticism about exclusively governmental regulation.

This survey's findings align closely with previous research on artificial intelligence in educational and research contexts. Earlier studies already recognized AI's potential in streamlining administrative tasks, facilitating adaptive learning methods, and enhancing personalized education—factors consistently noted even prior to the widespread availability of generative AI [77]. Similar to results presented here, earlier research has highlighted AI's beneficial role in automating text-based academic tasks such as content generation and manuscript preparation, underlining substantial perceived

utility among academics [78,79]. Additionally, respondents' concerns regarding misinformation and the fabrication of credible yet inaccurate bibliographic citations confirm issues previously documented in empirical assessments of generative AI tools, where substantive citation errors and misinformation have presented persistent challenges [51,52]. Moreover, consistent with previous survey-based studies, academic scientists demonstrate cautious optimism toward generative AI, balancing enthusiasm for potential efficiency improvements against the risks identified related to academic integrity and ethical standards in scholarly publishing [52]. This continuity with established scholarship reflects that the perceptions documented among STEM academic scientists reflect broader, ongoing dialogues around ethical and practical risks inherent to AI integration.

Notably, this survey also reveals important divergences from prior studies, particularly those involving student populations. Previous literature indicated predominantly positive student attitudes towards generative AI, emphasizing its capacity to improve academic engagement and personalized learning [29]. In contrast, academic scientists surveyed here report comparatively lower rates of generative AI use in activities involving direct interpersonal interaction or subjective evaluation, such as mentoring (17%), grading (11%), or conducting examinations (8%), indicating a selective rather than comprehensive adoption pattern. This divergence suggests a critical distinction between student enthusiasm—driven by immediate educational benefits—and the more cautious, responsibility-driven attitudes of academic scientists, who must balance innovation with preserving academic rigor and ethical standards. Furthermore, unlike earlier optimistic predictions regarding widespread chatbot adoption, which emphasized educational benefits such as personalized learning experiences, skill development, and substantial pedagogical support for educators [80,81], this survey indicates more selective adoption among academics, concentrated predominantly on content creation, and scholarly communication tasks rather than comprehensive integration across all academic functions.

While the study emphasizes misinformation concerns, other ethical implications warrant deeper exploration. The inherent bias in generative AI models, stemming from biased training data, significantly influences academic use by potentially perpetuating discriminatory outcomes, thus affecting fairness and equality within educational contexts [38]. Data privacy concerns also merit critical attention, especially regarding generative AI's role in processing and sharing sensitive academic and research data. For instance, AI-driven qualitative analysis methods raise important questions about ethical standards for data management [58]. Additionally, generative AI poses significant challenges for plagiarism detection, as it can produce highly sophisticated academic content that is nearly indistinguishable from human writing [82]. This limits the effectiveness of traditional detection tools and highlights the need for more advanced identification methods or a reevaluation of current academic integrity standards.

Regarding generative AI regulation, the study identifies a complex regulatory landscape perceived by academic scientists, who strongly favor (88) an academia-centered regulatory approach, primarily led by academic institutions, with complementary oversight by national professional associations (74%), journal editors (65%), and publishers (61%). Although a majority (71%) of respondents support some federal involvement, their preferences clearly emphasize cautious, evidence-based policies rather than outright governmental bans. Experimental survey data explicitly indicate that academic scientists' regulatory positions vary depending on how AI-related risks are presented, reflecting context-dependent rather than absolute regulatory stances. Current regulatory practices already align with these academic preferences. Universities are actively developing internal AI governance frameworks [53], academic publishers have begun establishing clear editorial standards for AI-generated content [83], and international organizations such as UNESCO have advanced global governance principles to shape national AI policies [84].

While this study provides valuable insights into the adoption intentions of academic scientists regarding generative AI, several limitations must be acknowledged. First, although based on probability sampling, the findings reflect responses from academic scientists across the six STEM disciplines included in the study, which may not represent the broader academic community. Additionally, as with any cross-sectional study, the findings reflect the period of data collection (late 2022–early 2023), capturing attitudes and experiences during the early adoption phase; perspectives may have since

evolved. Nevertheless, the findings of the paper on early-stage adoption behavior and perceptions provide a substantive and valuable basis for informing next stage research design and data collection. Smaller percentage differences should be understood within the survey's overall margin of error (±6.4%), though this does not affect the general interpretation of results, which point to trends in academic scientists' attitudes and experiences. Although no significant differences were observed in responses by gender or academic rank, the final sample does show variation in disciplinary representation. Civil and environmental engineering was more represented, while biology and public health were less so. These differences may reflect varying levels of exposure or familiarity with generative AI across disciplines.

Furthermore, although the survey did not inquire about other demographic characteristics, it did ask about gender, allowing us to examine data by gender. This is a notable limitation given that generative AI adoption and its impacts might vary across different demographic groups. The lack of race and ethnicity data is significant, especially considering how certain groups have faced disproportionate challenges within academia [85–88]. Additionally, while the survey examined generative AI's impact on the work of academic scientists, it did not address experiences with generative AI hallucinations or the generation of inaccurate information, which is an emerging concern. Indeed, clear evidence of misuse and malpractice already exists. Generative AI has generated high-quality fraudulent papers that are difficult to detect [82], and it can produce factually incorrect responses, including fabricated bibliographic citations [51,89]. These limitations suggest that future research should explore generative AI's effects across a wider range of academic disciplines and demographic groups, and more thoroughly investigate issues of accuracy and the reliability of AI-generated content. Such research is crucial for developing a comprehensive understanding of generative AI's role in academia and ensuring its responsible integration.

## Supporting information

**S1 Appendix. Non-response bias analysis.**
(DOCX)

**S2 Appendix. Number of randomly selected institutions for sampling scientists.**
(DOCX)

**S3 Appendix. Survey instrument: Generative AI uses and impacts in academic research and education.**
(DOCX)

## Author contributions

**Conceptualization:** Eric Welch.

**Data curation:** Jinghuan Ma, Tipeng Chen, Shaika Islam, Lesley Michalegko.

**Formal analysis:** Wenceslao Arroyo-Machado, Jinghuan Ma, Tipeng Chen, Shaika Islam.

**Funding acquisition:** Eric Welch.

**Methodology:** Wenceslao Arroyo-Machado, Timothy P. Johnson.

**Project administration:** Wenceslao Arroyo-Machado.

**Supervision:** Timothy P. Johnson, Eric Welch.

**Validation:** Timothy P. Johnson, Eric Welch.

**Visualization:** Jinghuan Ma, Tipeng Chen, Shaika Islam.

**Writing – original draft:** Wenceslao Arroyo-Machado, Jinghuan Ma, Tipeng Chen.

**Writing – review & editing:** Timothy P. Johnson, Shaika Islam, Lesley Michalegko, Eric Welch.

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
