## [Decision Letter · Decision Letter 0]

10 Feb 2025

Dear Dr. Arroyo-Machado,

Thank you for submitting your manuscript to PLOS ONE. After careful consideration, we feel that it has merit but does not fully meet PLOS ONE’s publication criteria as it currently stands. Therefore, we invite you to submit a revised version of the manuscript that addresses the points raised during the review process.

We look forward to receiving your revised manuscript.

Kind regards,

Kingsley Okoye

Academic Editor

PLOS ONE

**Journal Requirements:**

Reviewers' comments:

Reviewer's Responses to Questions

**Comments to the Author**

1. Is the manuscript technically sound, and do the data support the conclusions?

Reviewer #1: Partly

Reviewer #2: Partly

2. Has the statistical analysis been performed appropriately and rigorously?

Reviewer #1: Yes

Reviewer #2: I Don't Know

3. Have the authors made all data underlying the findings in their manuscript fully available?

Reviewer #1: Yes

Reviewer #2: Yes

4. Is the manuscript presented in an intelligible fashion and written in standard English?

Reviewer #1: Yes

Reviewer #2: Yes

**Reviewer #1:**  Review PLOS1

Overall

The authors present an interesting manuscript exploring the attitudes and experiences of academic scientific researchers toward generative AI. The article provides insightful and valuable contributions to this emerging area of study. However, many aspects require further clarification and refinement to enhance the manuscript's overall readability and impact.

Abstract

• How many respondents has the study included?

• Considering the results you wrote in your abstract, what do you mean by the term cautious?

• STEM participants only: how did this influence your results?

Introduction

• What is particularly evident in the STEM fields? Does the field of AI really cause the growing number of students in this field? (Line 60)

• You did not define generative AI. Only that it can produce human-like texts. But generative AI is more than that, right?

• This article explores the early adoption (Line 74). Why do you specifically mention the term early adoption, and not just adoption?

• In section 4… (line 95): does this section contain the results of the scoping review and the survey?

Literature review

• Could you clarify how the literature review was conducted? Specifically, what search strings did you use, and what were the criteria for determining an article's eligibility for inclusion?

• References 14 and 17 should be placed at the end of their respective sentences for clarity and alignment with academic citation standards.

• In the statement, "Academic libraries have utilised AI to optimise their services to enhance accessibility and efficiency" (Line 106), could you elaborate on what specific services you are referring to?

• Why is automated grading categorized as generative AI rather than simply AI? (Line 119)

• You wrote: "AI's potential to undermine creativity, hinder academic integrity, and exacerbate existing inequalities." How does AI contribute to exacerbating inequalities? Providing examples or further explanation would clarify this point.

• A reference is missing after the statement: "The unique capabilities and widespread adoption of generative AI have broadened and intensified these concerns."

• In the third paragraph of the literature review, there appears to be inconsistent use of the terms "AI" and "generative AI." Could you clarify whether these terms are being used interchangeably or if there is a specific reason for the distinction?

• You wrote: "... is balanced by concerns over risks such as misinformation, plagiarism, and falsified research. For example, while generative AI can produce high-quality text suitable for publication, it has limited capacity for research design and data analysis" (Line 159). How does this example relate to the concerns mentioned, such as misinformation, plagiarism, and falsified research?

• What does "the previous review” mean in Line 168?

• In "... benefits and challenges of generative AI" (Line 168), if you are referring to the previous paragraph, it seems the text pertains to AI in general rather than generative AI specifically.

• You wrote: "These include plagiarism, misinformation, and misidentification of AI-produced content as human work." and "For instance, in medical education, while there is optimism about AI's potential to reduce administrative burdens and enhance diagnostic accuracy, concerns persist about data privacy, the loss of personal contact with patients, and the adequacy of current AI technologies remain" (Lines 174-181). While your example in medical education provides meaningful insights, it does not directly align with the specific issues highlighted in the first sentence, such as plagiarism, misinformation, and the misidentification of AI-produced content. To ensure coherence, consider revising the example to better reflect these concerns or modifying the first sentence to encompass broader challenges associated with AI applications.

• Why do you not focus on generative AI rather then AI in generall in alinea 8 (Line 198-208), as your research is also only focused on generative AI.

• I think the following sentence: To bridge this gap (line 214), should be subsentence.

Material and methods

• Most important: the study can not be repeated. The reader has no idea what the survey entailed. Info is missing. Provide the exact questions and corresponding answer options that were explicitly asked during the survey for reproducibility.

• When mentioning SciOPS for the first time, write out its full name and provide a brief explanation of what it is, instead of just using the abbreviation. This helps readers unfamiliar with the term understand its relevance.

• The reference style for https://www.sci-ops.org/ is incorrect. Ensure it follows the required citation format for the journal or conference submission guidelines.

• The sentence on line 232 repeats the list of STEM fields: "six fields of STEM academic scientists biology, chemistry, civil and environmental engineering, computer and information science engineering, geography, and public health." Simplify it or rephrase it to avoid redundancy.

• The term "AAPOR" should be fully spelt out and briefly explained when first mentioned. For example, "AAPOR (American Association for Public Opinion Research)" if that is the intended meaning.

• The use of "etc." on line 239 is vague. Specify what additional items or categories it includes to provide clarity.

• These lines appear to be part of the results section. Move them to the appropriate section to maintain logical flow and structure.

• For clarity and compliance, include the Institutional Review Board (IRB) approval number or ID after the sentence: "Both surveys were approved by the Institutional Review Board at Arizona State University."

• The sentence starting with "The results indicate..." (lines 275–278) belongs in the results section rather than its current location. Relocate it to ensure consistency and proper organisation.

Results

• All in all, there are several sentences that seem to refer to opinions/conclusions rather then on findings.

• Ensure consistent use of "generative AI" and "AI" throughout the results section. Often, "AI" is used where "generative AI" would be more appropriate.

• The results would benefit from an analysis of differences in attitudes and concerns regarding generative AI among academic scientists across various STEM disciplines. Could you explore and include this aspect in your findings?

• The sentence, "Building on the understanding that generative AI will significantly impact daily life in the United States" (line 300), is an assumption and should not appear in the results section. Replace it with a neutral, evidence-based statement or move it to the discussion section if it is necessary for interpretation.

• Define what qualifies someone as an "early adopter" of AI. Specify the criteria or timeframe, as the current mention in line 318 leaves this unclear.

• In the sentence, "20% of academic scientists report using AI in both teaching and research, illustrating a robust integration into the academic environment" (line 321), consider whether "robust" is the right term.

• The statement, "Yet, there is a small segment, 20%, considering its future use, with 17% likely and 3% certainly planning to incorporate AI into their teaching," is unclear. Specify 17% and 3% of what. This follows a prior sentence mentioning 80% expressing reservations, which makes the proportions ambiguous. Rephrase for clarity and logical flow.

• You stated: "where 40% of users find value" (Line 358) and earlier wrote: "a substantial 40% are employing AI specifically within the domains of teaching and research" (Line 317). Is this referring to the same 40% of participants, or does the latter imply that 40% of the earlier 40% find value in AI? Please clarify this distinction for consistency.

• You wrote: "This distribution of responsibilities suggests a preference for an academic-led approach to AI regulation, complemented by governmental and international oversight, to effectively address the intricacies of AI's integration into academic settings." (Line 393-396). This is an insightful observation, but it remains abstract. Could you elaborate on what specific actions or frameworks academics, governments, or international bodies could implement to address these intricacies? Adding 1-2 sentences in the discussion section would strengthen your argument.

• You wrote: "revealing that the way questions are framed significantly influences their responses." (Line 414) It seems the responses were not influenced by how questions were framed but rather by the way answers were provided. This distinction is important. Please clarify the statement accordingly.

• In your methodology, you wrote: "SciOPS panel members were recruited from a full sample frame that includes around 18,500 randomly selected PhD-level academic scientists employed by Carnegie-designated Research Extensive and Intensive (R1) universities across the US with appointments in six fields of STEM academic scientists – biology, chemistry, civil and environmental engineering, computer and information science engineering, geography, and public health." However, in the results section (Line 440), you stated that findings are limited to biologists, biochemists, and civil & environmental engineers. Why were computer engineers and geography scientists excluded from the analysis? Please explain and ensure this is clarified in the text.

• You mentioned: "while the survey presents data by gender, it did not inquire about race or ethnicity, preventing an assessment of differences across these dimensions." (Line 442). This is a valid limitation. However, did you also consider assessing variations across scientific professions within STEM disciplines? Since this is a study about generative AI in academia, understanding differences across various scientific fields could provide valuable insights. Please address this or justify its omission.

• You used the term "AI-generated content" (Line 450). Is this terminology distinct from "Generative AI"? If so, please clarify the distinction in the text to ensure consistency and precision in the use of terminology.

Conclusion

R.485: by whom should this be done?

A real in depth discussion is missing.

NB.

The data are ‘available’. Is this made anonymous? How is this done?

**Reviewer #2: ** The manuscript presents an insightful study on the integration of Generative AI in academia, focusing on US academic scientists’ adoption, perceptions, and regulatory concerns. The topic is highly relevant, and the study provides valuable data from a nationally representative survey of STEM faculty. The findings contribute to ongoing discussions on AI’s role in education and research. However, certain areas require further clarification and expansion, particularly regarding methodology, theoretical grounding, and the discussion of ethical concerns. Below are specific strengths and areas for improvement.

Areas for Improvement:

• The manuscript would benefit from engagement with established technology adoption models, such as the Technology Acceptance Model (TAM) or Unified Theory of Acceptance and Use of Technology (UTAUT).

• Integrating these frameworks would help contextualize the faculty’s perceptions and concerns regarding AI adoption.

• While the study is based on a nationally representative survey, key methodological details are missing, including:

Sampling technique and representativeness of different institution types.

Survey response rate and potential response bias.

Validation of survey questions to ensure reliability and relevance.

Providing more transparency in these areas would enhance the study’s rigor.

• While the study mentions misinformation concerns (78%), it could also discuss other ethical implications, such as:

Bias in AI models and how it affects academic use.

Data privacy issues in AI-generated content.

Plagiarism detection challenges in AI-assisted research.

• A deeper engagement with these ethical dimensions would strengthen the manuscript’s contribution.

• The study focuses on STEM faculty, which may limit generalizability to the humanities, social sciences, and international academic institutions.

• Expanding the discussion on how AI adoption may differ across disciplines would enhance the broader relevance of the findings.

• While the study states that faculty support academic-led AI regulation, it does not provide concrete policy recommendations or engagement with existing AI governance frameworks.

• Expanding this discussion with references to current AI policies in education would improve the impact of the study’s findings.

**Do you want your identity to be public for this peer review?** For information about this choice, including consent withdrawal, please see our Privacy Policy

Reviewer #1: **Yes: ** Dr. Sil Aarts & Dirk Steijger, MSc

Reviewer #2: **Yes: ** Kamran Aziz earned his M.S. degree in Computer Science and Technology from Nanjing University of Information Technology, Nanjing, China. He is presently pursuing his Ph.D. in Cyberspace Security at the School of Cyber Science and Engineering, Wuhan University, China. Kamran is an expert in Natural Language Processing (NLP) and focuses on cutting-edge applications such as Fake News Detection, Named Entity Recognition, Sentiment Analysis, and Data Summarization & Augmentation. His research aims to enhance the reliability and efficiency of information processing in digital media.

---

## [Author Response · Author response to Decision Letter 1]

28 Mar 2025

"The responses to the reviewers' comments are included in the attached PDF file.

---

## [Decision Letter · Decision Letter 1]

5 May 2025

Dear Dr. Arroyo-Machado,

We look forward to receiving your revised manuscript.

Kind regards,

Kingsley Okoye

Academic Editor

PLOS ONE

Journal Requirements:

Additional Editor Comments :

We have now completed the review of your manuscript. The reviewers has suggested some areas for improvement in the manuscript. Please upload a revised manuscript and point by point response to the reviewers' comments.

Reviewers' comments:

Reviewer's Responses to Questions

**Comments to the Author**

Reviewer #3: All comments have been addressed

Reviewer #4: All comments have been addressed

Reviewer #5: (No Response)

2. Is the manuscript technically sound, and do the data support the conclusions?

Reviewer #3: Yes

Reviewer #4: Partly

Reviewer #5: No

3. Has the statistical analysis been performed appropriately and rigorously?

Reviewer #3: Yes

Reviewer #4: No

Reviewer #5: No

4. Have the authors made all data underlying the findings in their manuscript fully available?

Reviewer #3: Yes

Reviewer #4: Yes

Reviewer #5: Yes

5. Is the manuscript presented in an intelligible fashion and written in standard English?

Reviewer #3: Yes

Reviewer #4: Yes

Reviewer #5: No

Reviewer #3: Grammatical Mistakes

"First, despite being based on probability sampling, the findings are restricted to biologists, biochemists, and civil & environmental engineers," states Line 440 on Page 19. names "biochemists" inaccurately because the study is about biology, not biochemistry. "Biologists, chemists, and civil & environmental engineers" is the correct phrase.

Inconsistencies in Style:

Use of "AI" vs. "generative AI": The term "AI" and "generative AI" are used interchangeably throughout the manuscript, however the differences are not always made clear. To fit with the study's aim, the sentence "A sizable 65% of academic scientists have personally used AI" on page 14, line 317, for instance, may be changed to "generative AI." Early in the article, think about standardising vocabulary or defining "AI" as "generative AI."

Overall Evaluation

The manuscript satisfies PLOS ONE's standards for clarity, accuracy, and lack of ambiguity since it is written in standard English and presented in an understandable manner. The material is understandable to the intended audience of academic scholars, the language is formal and precise, and the structure makes sense. The majority of the text is free of significant typographical or grammatical problems, and the few that are found are small and simple to fix when it is revised.

Reviewer #4: The survey conducted here is on a very small sample, making this manuscript fewly relevant but this is compensated by a deep and intructing litterature review that led me to recommend the acceptance. Furthermore, the authors took a great caution in answering the concerns of the first round of reviewing, gicing more value to the manuscript.

In answered the data were available but it is still subject to approval and not available throught the article now.

I would nevertheless make the following suggestions :

- Line 33 : "Results indicate that 65% of scientists" I would rather say that "Results indicate that 65% of respondants".

- Line 210 : I am not sure the "Chubb et al." is relevant.

- Line 253 & 559 & 562 :Science told me there is only one human race, thus differencing race and ethnicity seems abusive to me.

- Line 280 : "The margin of sampling error for the AI survey estimates is +/-6.4 percentage points". The authors should state that this is for a population of 18 500 and a 95% CI.

- Line 317 : "The potential effects of climate change also register as a major point of academic concern, albeit slightly less so than generative AI, indicating a sense of urgency regarding environmental shifts." But these are within the margin of error.

- Line 448 : "Participants were randomly assigned one of four variations of the question". It would seem interesting to provide the margin of error for these sub-groups to the readers.

- Line 483 : Generative AI is predominantly used in educational settings for developing pedagogical materials (51%) and in research primarily for writing, reviewing, and editing tasks(40%)" but it only seems since when margin of erreors are taken into account this can be untrue.

I could also discuss the following the limits of this article :

- Sample size : The small sample size leads to an important margin of errors but more importantly reduces the relavance or significance of sub-groups analysis.

- Not all STEM widely represented (Geographs and civil engineers representing 66% of the cohort). This can both influence the result and the value of the data presented here and should be discussse omre in depth, since geographs seem to be more early adopters in bibliography. The signifiance upon other fields might also be discussed (For example, Biology with 16 answers might have very different behaviours that would not be highlighted by the present studies. Bibliography might also help on this point).

- It would also be interesting exploring deeper the results depending on the status of the respondants, whether professors, associates, or doctorants.

Reviewer #5: As a first-time reviewer for a manuscript that has gone through a previous round of review, I have approached my task by looking only at the latest revised version of the manuscript, to see how it reads. And, in doing so, I find the manuscript to be generally tackling an interesting question, but with various weaknesses in the way it does so that make it unclear to me what, precisely, the authors are trying to achieve with their paper. Briefly, my concerns consist of the following:

1. The manuscript is - in parts - in need of copy-editing, as there are several grammatical errors or unnatural linguistic phrases/constructions. Having a native speaker take a (possibly closer) look at it to catch these issues would likely help. Personally, I also find the use of various "power-adjectives" to be not only (occasionally) over the top and less suited for the academic context (e.g. that AI has had a "transformative impact" or represents "a significant leap forward" - both highly contestable - and contested - claims), but also occasionally a bit too reminiscent of the sort of language that chatbots tend to produce. As much as I can appreciate the subtle meta-commentary of using occasional chatbot-like phrases in a paper on the use of chatbots in academia, if this is intentional I think it should be flagged a bit more transparently, and if it's accidental then it's effect is a bit of an unintentional (partial) undermining of the expected academic tone. Either way, I think it requires some reconsideration.

2. I find the literature review lacking. Both in the sense that there are what seem to me to be serious gaps in the referenced literature (e.g. not including Gebru's work on "stochastic parrots" - although it's both crucial to the area covered and has had a massive impact on the subsequent discussion - seems strange to me). But also in the sense that there seems to be two rather larger structural issues in the section.

First, instead of discussing the benefits and risks of (gen)AI in academia, the authors go back and forth between the two, which makes for an awkward reading experience of "this sounds great, this sounds problematic, this sounds great, this sounds problematic". To be more concrete, the section has, after a brief introduction to AI in academia, a paragraph that starts with "However, despite perceived usefulness and improved ease of communication [33], there is insufficient understanding of generative AI’s long-term impacts on educational practices...". This is then followed by a paragraph starting with "As generative AI continues to transform education, its impact is equally profound in the field of research, revolutionizing research by potentially acting as "an autonomous researcher," transforming the way knowledge is produced across the research process", which is problematic both in its repetition of terms such as "transformative" and in its overreliance on "power"-adjectives as per point 1 above. But this is then, AGAIN, followed by a paragraph introducing various problems, beginning with the sentence "While many of the applications of generative AI to research activities appear promising, challenges and risks still loom large". This really needs some reconsideration of the structure of the themes and elements the authors wish to cover.

Second, and even more concerning, I don't find that the literature review section sufficiently sets up and qualifies the researchers' main focus, in the sense of clearly and unambiguously pointing to any important knowledge gaps. In the Introduction, the authors mention "Innovation Diffusion Theory", giving the impression this will constitute the theoretical framework for the remainder of the article, while the Lit Rev section presents this theory merely as one among others, with no indication as to why it constitutes a more relevant approach to their particular empirical undertaking. Further, the reliance on a rather broad range of references engenders some tension with the notion that the authors' own work has a sufficiently relevant value to warrant the research in the first place. I will say that I think this is primarily a writing issue: I believe that the research the authors have done is sufficiently important to warrant some attention and, ultimately, publication in a peer-reviewed journal. I just don't think that it is currently being set up in a sufficiently effective manner in the current version of their manuscript.

3. While I appreciate that the authors have based their work on the results of responses collected in '22 and '23, I think they need to more explicitly address the limitations this introduces, not least in terms of coming very short on the heels of the public launch of ChatGPT, meaning that attitudes today - in 2025, with all the far more wide-ranging experiences most of us have with genAI by now - might be significantly different. This is problematic in e.g. how the results are presented - phrasings such as "The data shows a majority engage with generative AI tools occasionally, reserving them for specific tasks rather than regular use" can be seen as potentially misleading - better replaced by introducing a qualifier such as "a majority during the studied time period" or similar - granted more clunky but at least more clearly indicating the actual knowledge that the responses allows one to surmise.

4. There are various issues with the statistics in the manuscript. For instance, the authors seek to assess "disciplinary variation" by performing a separate one-way ANOVA "across each response to each survey item (n=81 total comparisons)". They note that no potentially significant differences survived Bonferroni correction (.05 / 81). But given that some of their disciplines contain very few responders (e.g. 4 only in Public Health, 16 in Biology, 31 in Chemistry, etc.), this rather harsh correction for a rather absurdly large number of multiple comparisons cannot really be considered to allow the derivation of much of any relevant conclusion. I appreciate that this is one minor step - to remove concern that there could be significant hidden disciplinary variations among their respondents - but its application suggests to me a rather rudimentary appreciation of how to apply inferential statistics to real-world data. Essentially, with only 4 and 16 participants representing Public Health and Biology, respectively, a one-way ANOVA is ALREADY problematic, but to then further do an 81-comparison Bonferroni correction in order to suggest that there is no meaningful disciplinary variation in responses is such a stretch as to be generally irrelevant. Note also that this is but one example. To take another example: including four variations in phrasing of a key question introduces similar concerns, not least if cross-cutting with such categorizations as the respondents' field, rank, origin,, etc. Basically, the small numbers of respondents found in each such categorization generally aren't large enough to permit much of any meaningful comparisons.

5. The fact that there are essentially only descriptive (mostly frequency) statistics here are surprising to me. As much as I am of the opinion that researchers ought to do more exploratory analysis in general (as per Tukey), this ought typically to be in the service of either grounding subsequent inferential analysis or - perhaps - generating hypotheses for further (future) inferential testing. If the authors see themselves as falling with the exploratory-data-analysis camp, this is laudable and valuable, but then it needs to be more explicitly flagged as such throughout. In its current state, the manuscript reads more like a rather straightforward confirmatory-statistics paper, introducing aims, testing them, and drawing conclusions on the basis of the results found. But if the authors do see themselves as performing more confirmatory-style hypothesis-testing, then they would need to explore both descriptive and inferential statistics to a much greater degree (e.g. through basic measures of central tendency and variability, and inferential tools such as linear mixed models or whatever else the authors consider most appropriate), and further include statistical power calculations and the like, given their limited number of respondents. In general, I'd even go so far as to say that the repeated use of the term "analysis" isn't quite warranted by the actual research, which performs little actual analysis beyond presenting frequencies (percentages) of forced-choice responses to survey items. In this last respect, much of the manuscript findings read more like an interesting news story than a full-fledged research analysis of collected data.

In all, while the topic is interesting and highly relevant, I see a need for quite major revisions before I think this would be ready to be published in PLOS One.

**Do you want your identity to be public for this peer review?** For information about this choice, including consent withdrawal, please see our Privacy Policy

Reviewer #3: **Yes: ** Prakash Chandra Kasera

Reviewer #4: No

Reviewer #5: **Yes: ** Oskar MacGregor

---

## [Author Response · Author response to Decision Letter 2]

20 Jun 2025

Detailed responses to all reviewers' comments are attached in a PDF.

---

## [Decision Letter · Decision Letter 2]

1 Aug 2025

Generative AI and Academic Scientists in US universities: Perception, Experience, and Adoption Intentions

PONE-D-24-54545R2

Dear Dr. Arroyo-Machado,

We’re pleased to inform you that your manuscript has been judged scientifically suitable for publication and will be formally accepted for publication once it meets all outstanding technical requirements.

Kind regards,

Kingsley Okoye

Academic Editor

PLOS ONE

Additional Editor Comments (optional):

Reviewers' comments:

Reviewer's Responses to Questions

**Comments to the Author**

Reviewer #3: All comments have been addressed

2. Is the manuscript technically sound, and do the data support the conclusions?

Reviewer #3: Yes

3. Has the statistical analysis been performed appropriately and rigorously?

Reviewer #3: Yes

4. Have the authors made all data underlying the findings in their manuscript fully available?

Reviewer #3: Yes

5. Is the manuscript presented in an intelligible fashion and written in standard English?

Reviewer #3: Yes

Reviewer #3: The revised version of the manuscript addresses all issues raised by the reviewers, in particular with a better focus on facts, literature coverage, statistical robustness and acknowledgement of limitations. It has been accepted for publication in PLOS ONE based on the technical soundness, clarity and mechanism of the reasoning and clarity of the data presented. I suggest accepting the paper to be published in PLOS ONE, as it is currently already admissible for the journal's readership and adds valuable knowledge about the adoption of generative AI in academia.

---

## [Editor Report · Acceptance letter]

PONE-D-24-54545R2

PLOS ONE

Dear Dr. Arroyo-Machado,

I'm pleased to inform you that your manuscript has been deemed suitable for publication in PLOS ONE. Congratulations! Your manuscript is now being handed over to our production team.

Kind regards,

on behalf of

Dr. Kingsley Okoye

Academic Editor

PLOS ONE